# Genetic variability of bioactive compounds and selection for nutraceutical quality in kola [*Cola nitida* (Vent) Schott. and Endl.]

Daniel Nyadanu[1]*, Samuel Tetteh Lowor[1], Abraham Akpertey[1], Dèdéou Apocalypse Tchokponhoué[2], Prince Pobee[1], Jerome Agbesi Dogbatse[1], Daniel Okyere[3], Frederick Amon-Armah[1], Micheal Brako-Marfo[1]

**1** Cocoa Research Institute of Ghana, New Tafo-Akim, Ghana, **2** Faculty of Agronomic Sciences, Laboratory of Genetics, Biotechnology and Seed Science, University of Abomey-Calvi, Abomey-Calavi, Republic of Benin, **3** Department of Crop and Soil Sciences, Kwame Nkrumah University of Science and Technology, Kumasi, Ghana

* dnyadanu@gmail.com

**Data Availability Statement:** All relevant data are within the manuscript and its Supporting information files.

## Abstract

*Cola nitida* known as Kola serves as flavouring ingredient in the food industry and is also of great importance during traditional rites in Africa. Despite the well-known pharmaceutical values of the species, efforts to develop improved varieties with enhanced nutraceutical quality is limited due to unavailability of information on variation of genotypes in bioactive compounds in the nuts. The objectives of this research were to evaluate 25 genotypes of kola for bioactive contents, determine relationship between nutritional and phenolic traits and to identify kola genotypes with good nutraceutical quality for use in developing improved varieties. The kola genotypes were established in the field using a randomized complete block design with three replicates. Nuts harvested from the blocks, were bulked and used to quantify soluble and insoluble sugars, total protein, moisture, ash, fats, pH, polyphenols, tannins and flavonoids using completely randomized design with three replicates in the laboratory. Data were analysed by combining Analysis of Variance, Kruskal-Wallis test, correlation test and multivariate analysis. Significant variations (P < 0.05) were observed among the kola genotypes for the bioactive traits evaluated. Phenolic traits were more heritable than nutritional traits. Although not significant (P > 0.05), correlation between nutritional and phenolic traits was negative, whereas correlations among nutritional traits were weak. On the contrary, significant and positive correlations (P < 0.05) were observed among phenolic traits. The hierarchical clustering analysis based on the traits evaluated grouped the 25 genotypes of kola evaluated into four clusters. Genotypes A12, JB4, JB19, JB36, P2-1b, and P2-1c were identified as potential parental lines for phenolic traits selection in kola whereas genotypes A10, Club, Atta1 and JB10 can be considered for soluble and insoluble sugar-rich variety development. These findings represent an important step towards improving nutritional and nutraceutical quality of kola nuts.

**Funding:** The author(s) received no specific funding for this work.

**Competing interests:** The authors have declared that no competing interests exist.

## Introduction

Kola is an important nut crop in Africa. It belongs to the family Malvaceae, subfamily Sterculioideae with over 140 species indigenous to the tropical rain forest of Africa [1–3]. *Cola nitida* (Vent) Schott. & Endl. and *Cola acuminata* (Beauvoir) Schott & Endl. are the commercially important species. *Cola nitida* is easily distinguished by its nuts of two cotyledons. *Cola acuminata* has three to six cotyledons. Outside mainland Africa, kola species has been introduced and largely grown in the tropical South and Central America and the West Indies [4–6].

The commercial product of kola is its nuts which are masticated to remain vigilant and to encourage salivation. The nuts are also used in products such as wine, chocolate and many beverages as flavouring agents. The kola nuts are nutritious and contain high levels of caffeine (2.8%), theobromine (0.05%) and phenolic compounds [7–9]. The nuts are also rich in amino acids. Glumatic acid and aspartic acid are of particular importance [10]. Glutamic acid is one of the few free amino acids occuring in appreciable concentration in the brain and plays the principal role in neuron transmission [11]. The aspartic acid helps to promote a robust body metabolism and it is used to treat depression and fatigue [12]. The kola industry offers a lot of employment and income opportunities to people involved in the harvesting, processing, packaging and transportation of nuts [13–15]. The crop has socio-cultural importance in Africa especially during traditional rites [16,17].

Kola is one of the prioritized indigenous fruit tree species for domestication and integration into farming systems in Africa to support nutritional and income generation to alleviate poverty among local people [18,19]. Currently, there is an increasing interest in kola as a major source of bioactive compounds. The fresh nuts of kola are high in phenolics and other essential bioactive compounds [20–22]. Bioactive compounds (phytonutrients) such as carotenoids and phenolic acids are health-promoting compounds that act against cardiovascular and various types of cancer [23]. Phenolic compounds exert a potent antioxidant activity and are analgesic, anti-carcinogenic, anti-diabetic, anti-inflamatory, anti-microbial, anti-obesity, cardioprotective, hypotensive and neuroprotective [24]. The presence of kolanin and theobromine makes the nuts of kola suitable for development of new pharmaceuticals and foods [25]. Also, the volatile oil from *C. nitida* exhibits antioxidant properties and involves in apoptosis and therefore has potential to be an important medicinal resource [26,27].

Despite the known importance of bioactive compouds content of kola, efforts to breed varieties with enhanced levels of these compounds is lacking. Breeding of fruits with enhanced amounts of nutritional and phenolic traits is deemed very necessary [28–31] to promote good health among consumers. Farmers and other stakeholders along the kola value chain in Ghana use kola for medicinal purposes and indicated their preference for kola varieties with high nutritional and medicinal compounds content [32]. Involving end-users preferences in goal setting and product development is highly recommended for success and adoption of new varieties [33–37]. Therefore in kola, it is very important to include this client-oriented trait in selection and breeding of improved varieties. Kola varieties that are rich in beneficial bio-active compounds and limited in anti-nutrient contents are desirable and have been the target of many breeding programmes [38–41].

As previously reported in other studies, bioactive compound contents of fruits are greatly influenced by the genetic background of crops [42–44]. To make much progress, large germplasm resources with high variations for these bioactive compounds are required. At the Cocoa Research Institute of Ghana, some germplasm of *Cola nitida* has been collected and conserved as field collections [45]. However, variation in the bioactive compounds content of these kola genotypes in Ghana and elsewhere in the world has not yet been documented. Also information on heritability, genetic advance and association among bioactive contents of kola

genotypes which are necessary to guide the breeding approach to use and to maximize selection efficiency have not been reported. Lack of these key pieces of information twarted identification of promising genotypes and breeding of improved varieties with nutraceutical quality nuts. Enhancement of bioactive compounds content of improved varieties as desired by clients require information on quantitative variation and diversity of kola genotypes for the bioactive traits. Quantification of genetic variation of cultivars is necessary for efficient use of plant genetic resources and for determination of relationship between desirable traits [46].

With the backdrop of the limitations above, this study was therefore carried out with the aim to (i) assess phenotypic variation in bioactive compounds among 25 genotypes of *C. nitida*, (ii) determine the relationship between nutritional and phenolic traits and (iii) identify kola genotypes with good nut qualities for use in developing improved varieties.

## Materials and methods

### Genetic materials and description of study area

Twenty-five (25) genotypes of *Cola nitida* originating from the Cocoa Research Institute of Ghana (CRIG) kola breeding program, were evaluated in this study. Table 1 shows the list of the genotypes evaluated and characteristics of their pod and nut yields. The fruits analyzed were harvested from field grown plants of each genotype conserved in kola germplasm collection (Plot MX2) at Tafo in the Eastern Region of Ghana. The MX2 kola collection was planted in July 1987. The evaluation for the bioactive compounds content of the kola genotypes was carried out from august 2018 to February 2019 during the harvest season of kola. CRIG is located at an altitude of 222 m above sea level. The weather conditions during 2018 and 2019 at the Cocoa Research Institute of Ghana, Tafo, where the germplasm collection is located is as shown in Fig 1. In 2018 the mean maximum temperature ranged from 29.52˚C to 34.61˚C. Average rainfall ranged from 0.00 mm to 18.69 mm. In 2019, mean maximum temperature ranged from 29.43˚C to 34.82˚C. Average rainfall was the highest in the month of June and least in December 2019. Mean daily sunshine ranged from 3.69 in July to 7.05 in April, 2019. The soil on which the germplasm is located is of sandy-loam type and its physicochemical properties are shown in Table 2.

### Experimental design and collection of kola pod samples

The 25 kola genotypes were established in the field using a randomized complete block design with three replicates in the year 1987 at Tafo on a 5.54 acre land. The spacing was 9.9 m x 9.9 m and five stands were planted per plot. Cultural practices such as mistletoe removal, pruning and weeding were applied on a reguar basis. Kola pods were randomly collected from the stands of each genotype per replication and bulked for biocompounds quantification.

### Analysis of kola nuts for bioactive compounds content

All the analysis of the nuts were initiated on the next day following the harvest of the pods.

### Determination of soluble and insoluble sugars content

Soluble and insoluble sugars in kola nuts were quantified following [48] method using phenol-sulphuric acid reagent [49].

### Extraction of alcohol soluble sugars

30 ml of 80% ethanol solution was added to 0.5 g of grounded kola nut sample and refluxed on a hot plate for 30 minutes. The solution was allowed to cool and the supernatant was decanted

**Table 1. Kola genotypes evaluated showing their sources and pod and nut yield.**

| Genotype | Source | Pod yield (Kg. ha$^{-1}$)[a] | Nut weight (g)[a] | Nut colour[a] |
|---|---|---|---|---|
| A1 | Asikam, E/R* | 597 | 2,272 | white |
| A10 | Asikam, E/R* | 355 | 1,753 | white |
| A12 | Asikam, E/R* | 252 | 1,158 | red |
| A2 | Asikam, E/R* | 690 | 2,431 | white |
| A22 | Asikam, E/R* | 967 | 1,757 | white |
| A26 | Asikam, E/R* | 1,055 | 2,529 | white |
| Atta1 | Tafo, E/R* | 41 | 752 | white |
| Club | Tafo, E/R* | 270 | 1,452 | white |
| JB1 | Juaben, A/R+ | 1,225 | 3,569 | white |
| JB10 | Juaben, A/R+ | 705 | 2,173 | red |
| JB17 | Juaben, A/R+ | 489 | 2,200 | white |
| JB19 | Juaben, A/R+ | 445 | 1,445 | pink |
| JB20 | Juaben, A/R+ | 1,238 | 3,402 | red |
| JB22 | Juaben, A/R+ | 834 | 2,973 | red |
| JB26 | Juaben, A/R+ | 352 | 1,231 | red |
| JB27 | Juaben, A/R+ | 399 | 1,805 | red |
| JB32 | Juaben, A/R+ | 276 | 2,595 | red |
| JB35 | Juaben, A/R+ | 952 | 2,292 | red, pink and white |
| JB36 | Juaben, A/R+ | 441 | 1,383 | red |
| JB37 | Juaben, A/R+ | 1,065 | 1,601 | red |
| JB4 | Juaben, A/R+ | 146 | 912 | red |
| JB40 | Juaben, A/R+ | 274 | 2,710 | White (big nuts) |
| JB9 | Juaben, A/R+ | 465 | 2,122 | red |
| P2-1b | Kade Okumani, E/R* | 674 | 2,048 | red, pink, white |
| P2-1c | Kade Okumani, E/R* | 768 | 1,958 | red, pink, white |

[a]Source: Cocoa Research Institute of Ghana Annual report [47].

* E/REastern region.

+A/RAshanti region.

into a separate receiver flask. This procedure was repeated three times. After this, all the filtrate was bulked and the ethanol was evaporated under reduced pressure using a rotary evaporator (BÜCHI 011 made in Switzerland EL 131). After the evaporation of ethanol, ethanol volatile, water and insoluble substances were precipitated with 0.3 N Barium Hydroxide Ba(OH)$_2$ solution and 5% Zinc Sulphate (ZnSO$_4$) solution and filtered into a clean flask using Whatman No. 54 filter paper. The filtrate was then passed through a mixture of Zeokard 225 (H$^+$), a cation exchange resin and Deacidite FF(OH) and filtered. The final volume of filtrate was recorded and kept in a falcon tube in a freezer at -80°C until analysis. A maximum of 1ml each of the extracts of alcohol soluble samples were taken into a test tube. 1ml of 10% phenol reagent was added to each sample and this was followed by 5ml of concentrated sulphuric acid. The mixture was then allowed to cool and absorbance was read at 490nm using the UV/ V spectrometer (Jenway 6405 UV/UV spectrophotometer). The standard calibration was prepared using glucose at concentrations 20, 40, 60, 80 and 100 ppm.

## Extraction of alcohol insoluble or acid soluble sugars

20 ml of 0.75 M sulphuric acid (H$_2$SO$_4$) was added to the residue in the flask and refluxed on a heater for one (1) hour. The solution was cooled and filtered. The acid filtrate was neutralized

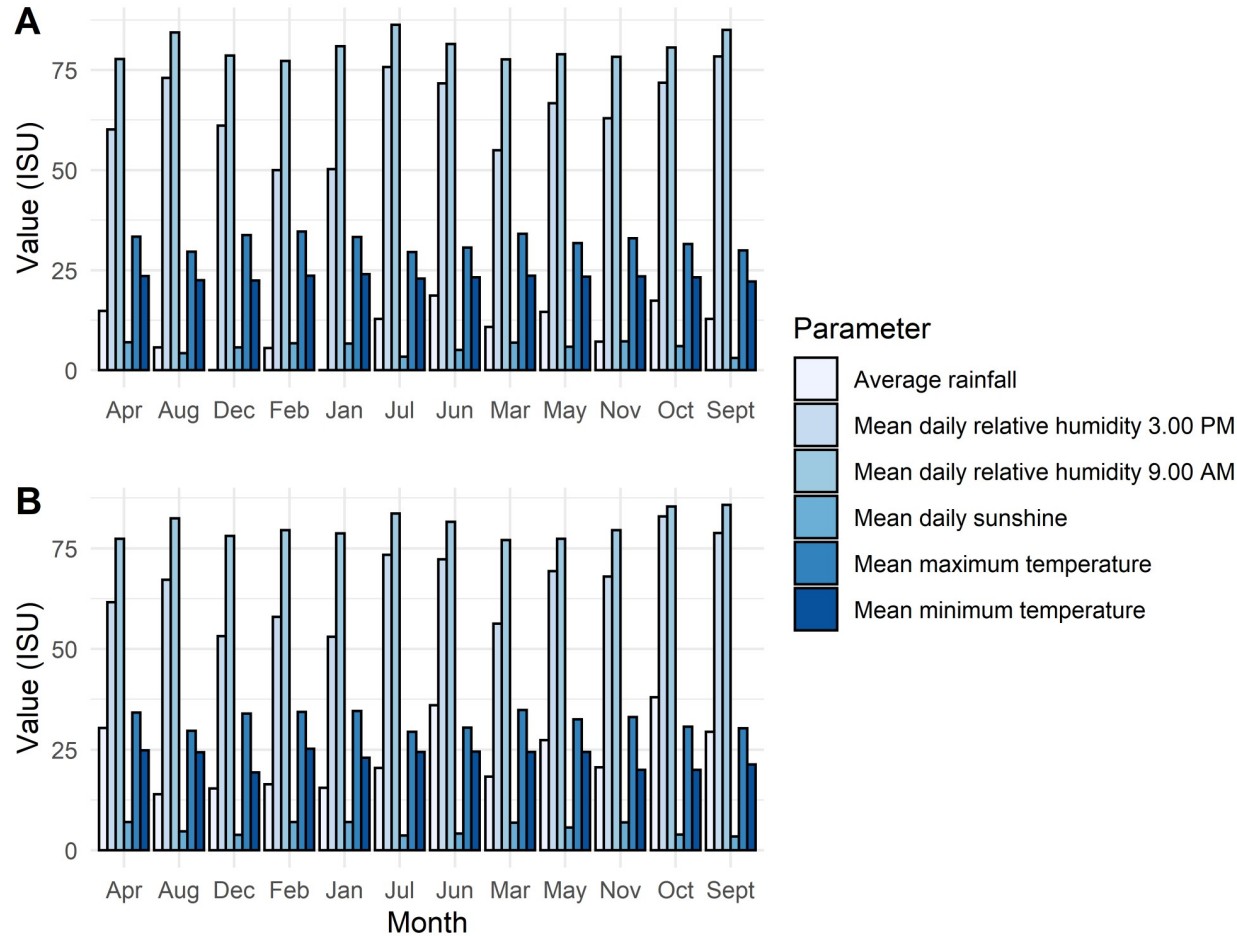

**Fig 1. Weather data of Tafo in 2018 (A) and 2019 (B) season, the period in which the study was carried out.**

**Table 2. Physicochemical properties of the soil on which the plants are grown.**

| | Depth (cm) | |
|---|---|---|
| **Property** | **0–15** | **15–30** |
| pH | 5.59 | 5.75 |
| Organic C (%) | 0.96 | 0.81 |
| Total N (%) | 0.11 | 0.09 |
| Available P (mg kg$^{-1}$) | 24.98 | 20.49 |
| Exchangeable K (cmol kg$^{-1}$) | 0.06 | 0.04 |
| Exchangeable Mg (cmol kg$^{-1}$) | 1.05 | 0.73 |
| Exchangeable Ca (cmol kg$^{-1}$) | 3.03 | 2.39 |
| Cu (µg/g) | 2.18 | 2.1 |
| Sand (%) | 72.04 | 72.44 |
| Silt (%) | 14.8 | 14.8 |
| Clay (%) | 13.16 | 12.76 |
| Textural class (USDA) | Sandy loam | Sandy loam |

with Barium carbonate ($BaCO_3$) to pH 7. The solution was then centrifuged at 1000rpm for 30 mins at 4°C. The filtrate was then decanted and cleared using 0.3N Barium Hydroxide (Ba $(OH)_2$) solution and 5% Zinc Sulphate (ZnSO4) solution and filtered into a clean flask using Whatman NO. 54 filter paper. The filtrate was then passed through a mixture of Zeokard 225 (H+), a cation exchange resin and Deacidite FF (OH), an anion exchange resin and filtered. The final volume of the filtrate was recorded and kept in a falcon tube in a freezer at -80°C until analysis. A maximum of 1ml each of the extracts of alcohol insoluble samples were taken into a test tube. 1ml of 10% phenol reagent was added to each sample and this was followed by 5ml of concentrated sulphuric acid. The mixture was then allowed to cool and absorbance was read at 490nm using the UV/V spectrophotometer (Jenway 6405 UV/UV spectrophotometer). The standard curve was prepared using glucose at concentrations 20, 40, 60, 80 and 100 ppm.

Total sugars/carbohydrate was taken as the sum of the alcohol soluble and the alcohol insoluble sugars.

## Phenols analysis

Folin-Ciocalteau colorimetric method [50] was used to determine the total phenolic content of the kola nut extracts. 30 ml of 80% acidified Methanol (Methanol: Conc. HCl = 79:1) was added to 0.2g of defatted kola nut sample in a 50 ml falcon tube and placed on a shaker for two (2) hours at 420 $min^{-1}$. After two hours, the extract was filtered. 1 mil of filtrate was then taken into a test tube and 5 ml of 1:9 ml of Follin-Ciocalteu's Phenol reagent was added to the content in the test tube. After 8 minutes, the reaction was neutralized by adding 4mL of 75 $gL^{-1}$ sodium carbonate and incubated for 1 hour at 30°C and 1 hour at 0°C. Absorbance was read at 760 nm using the UV/V spectrophotometer (Jenway 6405). Readings were calibrated using a catechin standard curve ranging from 0 to 100 ppm.

## Flavonoids analysis

The total flavonoid content was determined by the aluminium chloride colorimetric method as previously described [51]. 1 ml of polyphenol extract was taken into a test tube and 600 μl of a 5% sodium nitrite ($NaNO_2$) solution was added to the content in the test tube and the mixture was allowed to stand for 6 minutes. 150 μl of 10% aluminium trichloride was then added and incubated for 5 min. This was followed by the addition of 750 μl of NaOH (1.0M) and the final solution was adjusted to a volume of 2500 μl with distilled water. Absorbance was read after 15 minutes of incubation at 510 nm using the spectrophotometer (Jenway 6405). Catechin was used as the standard.

## Total condensed tannins content analysis

Tannins were assayed using the procedure of Price et al. [52]. 5 ml of vanillin/HCL reagent (0.5 g vanillin in 4% Hydrochloric acid in methanol (v/v) and 1.5 mil of concentrated Hydrochloric acid) was added to 1 ml of polyphenol extract. The mixture was incubated in the dark for 15 minutes and absorbance was read at 500nm. The standard used was catechin.

## Determination of nitrogen and total protein content by Kheldahl method

Nitrogen (N) was extracted and analyzed by the digestion of kola nuts using the micro-kjeldahl method as described by [53]. 2.5 g of air-dried kola nut samples were weighed into digestion tubes. 0.5g of Catalyst (1:5:25g Selenium (Se), Copper Sulphate ($CuSO_4$), Potassium Sulphate ($K_2SO_4$) ratio) was added. 12 ml of concentrated nitrogen free sulphuric acid were added to the samples and digested for 2 hours at 350°C. The digested samples in the tubes were allowed

to cool in a fume chamber until there were no fumes evolving. The digest was washed and the tubes rinsed about three times with distilled water into bigger tubes for digestion. The distilled samples (distillates) which contained the ammonia compounds were then collected in receiver flasks and titrated with standardized 0.02N sulphuric acid. The percentage nitrogen in the sample was then calculated using the formula below:

$$\% \, N = (Titre \, value \, of \, sample \times Normality \, of \, Acid \times 1.401)/Weight \, of \, sample \, (g) \quad (1)$$

$$Protein = Protein = N \times 6.25 \quad (2)$$

## Statistical analyses

**Descriptive statistics and variation in bioactive compounds content among the kola genotypes.** Data collected on bioactive contents of kola nuts were summarized using descriptive statistics (e.g. average, coefficient of variation, skewness, kurtosis). Difference among genotypes for the 10 nutritional and phenolic traits measured was tested by means of analysis of variance (ANOVA) or Kruskal-Wallis test where appropriate.

**Estimates of genetic parameters of the nutritional and phenolic traits.** Estimation of genetic and phenotypic coefficients of variation, expected genetic advance/genetic gain, as well as percentage of genetic advance were carried out using the functions provided by Farshadfar et al. [54]:

$$\delta^2_e = MS_e \quad (3)$$

$$\delta^2_g = (MS_g - MSe)/r \quad (4)$$

$$\delta^2_p = \delta^2_g + \delta^2_e \quad (5)$$

$$PCV(\%) = \frac{\sqrt{\delta^2_p}}{\bar{x}} \times 100 \quad (6)$$

$$GCV(\%) = \frac{\sqrt{\delta^2_g}}{\bar{x}} \times 100 \quad (7)$$

$$ECV(\%) = \frac{\sqrt{\delta^2_e}}{\bar{x}} \times 100 \quad (8)$$

$$H^2 = \frac{\delta^2_g}{\delta^2_p} \times 100 \quad (9)$$

$$GG = \left( \frac{i.\delta^2_g}{\sqrt{\delta^2_p}} \right) \times \frac{100}{\bar{x}} \quad (10)$$

$$GG(\%) = GA \times \frac{100}{\bar{x}} \quad (11)$$

where σ²e = environmental variation, Mse = error mean square, Msg = genotype mean square, Vg = genetic variation, r = number of replication, X̄ = Mean, VP = phenotypic variation, σ²g = genetic variance, σ²p = phenotypic variance, PCV = phenotypic coefficient of variance, GCV = genotypic CV, H² = broad sense heritability, GG = genetic gain, GG (%) = percentage of genetic gain, the standard selection differentials (i) for 5% selection intensity was 2.06.

**Relationship between nutritional and phenolic traits.** The relationships between nutritional and phenolic traits were established and tested for their significance using Pearson and Spearman correlation tests. A principal component analysis was carried out on the 10 traits of the study to identify the most meaningful components using the *PCA ()* function of the FactoMineR package [55]. A hierarchical cluster analysis was performed on the principal components retained to group the genotypes based on their similarities using the *HCPC ()* function of the same statistical package. Graphical outputs of the multivariate analysis were plotted using the *fviz ()* function of the factoextra package [56]. All the analyses were performed using the R environment Version (3.6.2) [57].

## Results

### Descriptive data and variation in bioactive compounds content

Quantitative variation of nutritional and phenolic traits among the 25 kola genotypes is shown in Table 3. Coefficients of variation (CV) for nutritional traits ranged from 20.95% for ash to 40.57% for total protein. The CV for phenolic traits ranged from 25.61% (polyphenols) to 38.93% (flavonoids). Insoluble and soluble sugars, flavonoids, pH, polyphenols, proteins and tannins were positively skewed. Ash, fat and moisture were negatively skewed. Kurtosis among the nutritional and phenolic traits was between -0.27 for fat (%) and 14.61 for pH.

There were significant (p<0.05) variations among the 25 kola genotypes for all nutritional and phenolic characters evaluated (Fig 2-1(A-F) and 2-2(G-J)). For instance, the soluble sugar content (df = 24, Kruskal-Wallis chi-squared = 57.08, Fig 2-1A) for genotype Atta1 is more than three fold higher than those observed for five other genotypes (JB9, JB27, A22, JB32, and JB22) and more than two fold higher than those of three other genotypes (JB37, A12 and JB20). Likewise, the genotype JB20 has a level of insoluble sugars that is more than twice higher than that of genotype P2-1b (df = 24, F = 12.51, Fig 2-1B). Similar trends of variation were also observed among genotypes for other traits such as Ash (df = 24, Kruskal-

**Table 3. Summary of descriptive statistics characterizing the 25 kola genotypes.**

| Variable | Min | Max | Mean | Range | Std Dev | CV | Skewness | Kurtosis |
|---|---|---|---|---|---|---|---|---|
| %Ash | 0.30 | 4.25 | 2.66 | 3.95 | 0.56 | 20.95 | -1.35 | 7.74 |
| %Fat | 0.15 | 0.79 | 0.47 | 0.69 | 0.15 | 31.43 | -0.02 | -0.27 |
| %Moisture | 48.17 | 68.06 | 58.76 | 19.89 | 4.50 | 7.66 | -0.30 | -0.40 |
| Insoluble sugar | 16.95 | 73.00 | 41.79 | 56.05 | 11.44 | 27.37 | 0.42 | 0.08 |
| Soluble sugar | 2.50 | 11.06 | 5.06 | 8.56 | 1.99 | 39.25 | 1.09 | 0.77 |
| Flavonoids | 1.12 | 6.52 | 3.24 | 5.40 | 1.26 | 38.93 | 0.76 | 0.02 |
| pH | 5.65 | 6.88 | 5.97 | 1.23 | 0.18 | 3.06 | 3.19 | 14.61 |
| Polyphenols | 22.30 | 66.1 | 38.46 | 43.8 | 9.85 | 25.61 | 0.49 | 0.15 |
| Proteins | 3.06 | 13.52 | 7.33 | 10.46 | 2.98 | 40.57 | 0.39 | -0.99 |
| Tannins | 17.85 | 72.51 | 44.1 | 54.66 | 16.16 | 36.64 | 0.21 | -1.24 |

Min = Minimum, Max = Maximum, Std Dev = Standard deviation, CV = Coefficient of variation.

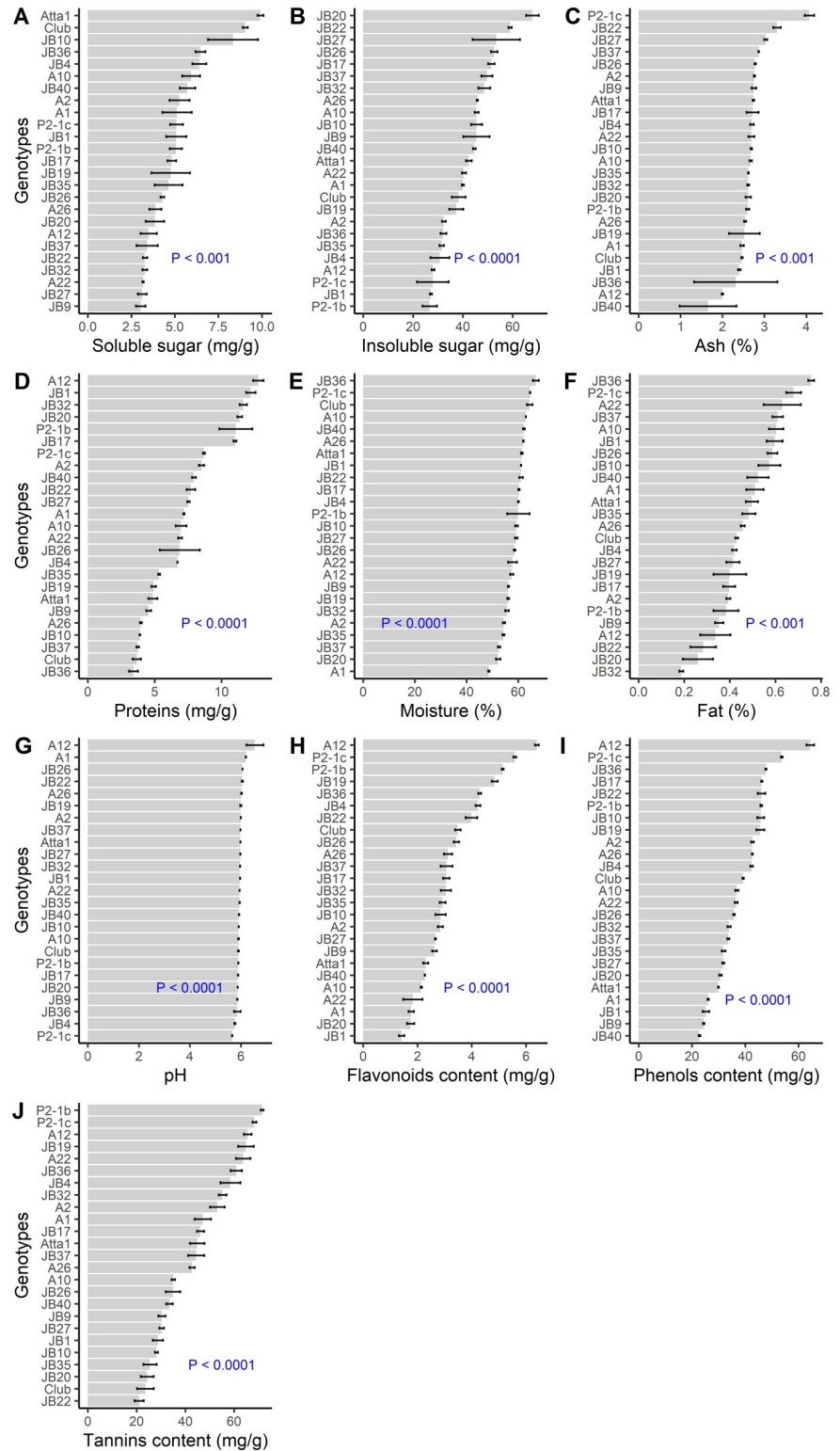

**Fig 2. Variation in nutritional and phenolic traits among the 25 kola genotypes.**

**Table 4. Variance components and estimates of genetic parameters for the ten bioactive compounds.** Please refer to Eqs 3–11 for the definition of the genetic parameters.

| Trait | $\delta^2 e$ | $\delta^2 g$ | $\delta^2 p$ | PCV% | GCV% | $H^2$ (%) | G.G. | G.G. (%) |
|---|---|---|---|---|---|---|---|---|
| %Ash | 0.19 | 0.118 | 0.308 | 20.84 | 12.89 | 38.31 | 0.44 | 16.45 |
| %Fat | 0.005 | 0.02 | 0.022 | 47.05 | 30.06 | 90.9 | 0.28 | 59.09 |
| %Moisture | 3.44 | 17.33 | 20.77 | 7.76 | 7.08 | 83.44 | 7.83 | 13.32 |
| Insoluble sugar | 28.37 | 105.83 | 134.2 | 27.72 | 24.62 | 78.86 | 18.82 | 45.03 |
| Soluble sugar | 0.92 | 3.11 | 4.03 | 39.68 | 63.28 | 77.15 | 3.205 | 63.34 |
| Flavonoids | 0.06 | 1.58 | 1.64 | 39.47 | 38.77 | 96.52 | 2.54 | 78.36 |
| pH | 0.016 | 0.019 | 0.036 | 3.18 | 2.37 | 55.43 | 0.22 | 3.62 |
| Polyphenols | 1.48 | 98.175 | 99.66 | 25.96 | 25.76 | 98.51 | 20.26 | 52.68 |
| Total protein | 0.6 | 8.49 | 9.09 | 41.11 | 39.72 | 93.36 | 0.58 | 7.91 |
| Tannins | 16.26 | 251.41 | 267.67 | 37.09 | 35.95 | 93.93 | 31.66 | 71.79 |

Wallis chi-squared = 56.22, Fig 2-1C), proteins (df = 24, Kruskal-Wallis chi-squared = 69.6, Fig 2-1D), phenols (df = 24, Kruskal-Wallis chi-squared = 72.71, Fig 2-2I) and tannins (df = 24, Kruskal-Wallis chi-squared = 70.25, Fig 2-2J). Noticeably, there was a four-fold variation in fat content (df = 24, F = 11.5, Fig 2-1F) between genotypes JB36 and JB32 and a nearly five-fold variation in flavonoids content between genotypes A12 and JB1 (df = 24, Kruskal-Wallis chi-squared = 71.05, Fig 2-2H). The differences among the genotypes were not apparent for the variables pH (df = 24, Kruskal-Wallis chi-squared = 63.38, Fig 2-2G) and moisture content (df = 24, F = 16.43, Fig 2-1E).

## Estimates of genetic parameters of the nutritional and phenolic traits

Environmental variance ($\delta^2 e$) ranged from 0.005 for fat to 28.37 for insoluble sugars (Table 4). Slight differences were observed between phenotypic coefficient of variation (PCV) and genotypic coefficient of variation (GCV) for majority of the traits except ash (%) and soluble sugars which indicated a wide difference between PCV and GCV. In the case of ash, PCV (%) was more than GCV (%) whereas for soluble sugars GCV (%) was higher than PCV (%). Phenotypic coefficient of variation varied from 3.18 for pH to 47.05 for fat (%) while GCV ranged from 2.37 for pH to 63.28 for soluble sugars (Table 4). Fat, insoluble sugars, soluble sugars, flavonoids, polyphenols and tannins had high heritability and high percentage genetic gain values (Table 4). In the case of ash and pH, lower values for heritability and genetic gain were observed. Total protein had high heritability value but a very low percentage genetic gain.

## Relationship between nutritional and phenolic traits measured for the 25 kola genotypes

Table 5 indicates that correlation coefficients among studied traits ranged from -0.001 (moisture and tannins) to 0.78 (flavonoids and polyphenols). Significant and positive correlations were observed among the phenolic traits, in particular between flavonoids and polyphenols (r = 0.78, P < 0.001). Correlations between nutritional traits were weak in general, except correlation between moisture content and soluble sugars. No significant correlations existed between nutrient and phenolic traits in general, except between insoluble sugars and tannins (r = -0.52, P < 0.05).

The principal component analysis indicated that PC1 and PC2 together explained above 50% of the total variation among the kola genotypes evaluated for the measured nutritional

**Table 5. Correlation matrix among the 10 bio-compound traits measured on 25 kola genotypes.**

|  | Proteins | Flavonoids | Tannins | Polyphenols | Ash | Fat | pH | Moisture | SS | IS |
|---|---|---|---|---|---|---|---|---|---|---|
| **Proteins** | X | 0.60 | 0.35 | 0.85 | 0.73 | **0.03** | 0.67 | 0.41 | 0.1 | 0.65 |
| **Flavonoids** | -0.11 | X | **0.02** | 3.84E-06 | 0.82 | 0.54 | 0.87 | 0.24 | 0.88 | 0.13 |
| **Tannins** | **0.19** | **0.45** | X | **0.02** | 0.64 | 0.63 | 0.87 | 0.99 | 0.83 | **0.007** |
| **Polyphenols** | **0.039** | **0.78** | **0.46** | X | 0.74 | 0.81 | 0.72 | 0.14 | 0.52 | 0.25 |
| **Ash** | -0.07 | **0.04** | -0.09 | **0.07** | X | 0.99 | 0.86 | 0.6 | 0.15 | **0.04** |
| **Fat** | -0.42 | -0.12 | **0.10** | -0.05 | **0.01** | X | 0.53 | 0.06 | 0.05 | 0.29 |
| **pH** | **0.08** | -0.03 | -0.03 | -0.07 | -0.03 | -0.13 | X | 0.07 | 0.20 | 0.34 |
| **Moisture** | -0.17 | **0.24** | -0.001 | **0.30** | -0.11 | **0.36** | -0.36 | X | **0.01** | 0.26 |
| **SS** | -0.27 | **0.03** | **0.04** | **0.13** | -0.29 | **0.38** | -0.26 | **0.50** | X | **0.04** |
| **IS** | -0.09 | -0.30 | -0.52 | -0.23 | **0.39** | -0.34 | **0.19** | -0.23 | -0.41 | X |

SS = Soluble sugars; IS = Insoluble sugars. Values at the lower diagonal represent coefficients of correlation calculated using the Pearson or Spearman methods (positive correlations are in bold). Values at the upper diagonal are probability values of the correlation test between paired variables (values in bold in the upper diagonal indicate significance at α = 5%).

and phenolic traits (Fig 3A). The eigenvector for PC1 was 28.62% and it was mainly defined by phenolic characters; flavonoids, polyphenols and tannins. The eigenvector for PC 2 was 23.51%. The PC2 was mostly explained by nutrient-related traits such as total proteins, soluble sugars and fats.

An analysis of the contribution of variables to the first two principal components indicated that variables such as flavonoids, polyphenols, tannins, moisture and fat gave above average to the variability in the first two dimensions (S1A Fig). Likewise, genotypes A12, P2-1c, JB 20, JB 36, P2-1b and JB 32 recorded contributions which were higher than the average for the variability in the first two components (S1B Fig and Fig 3B).

The dendrogram grouped the 25 kola genotypes into four clusters. The genotypes that constitute membership to these four clusters are presented in Fig 3C. Individuals from cluster 1 (C1) were characterized by an insoluble sugars content which was higher than the average for all the genotypes. Tannins, moisture, soluble sugars, phenols and flavonoids contents were extremely lower than the average of all the genotypes. Cluster 2 (C2) was exlusively characterized by its higher value of soluble sugars content compared to the average for all the genotypes. Kola genotypes that constituted the third cluster (C3) were characterized by flavonoids, phenols and tannins contents. The contents of these phenolic compounds of individuals in C3 were higher than the average for all the genotypes. The individuals in C3 were however lower in pH and insoluble sugars content as compared to the average for all the 25 kola genotypes. A12 was the only member of the cluster 4 (C4). This genotype was markedly distinguished from the other genotypes by its pH, phenols and flavonoids content which were higher compared to the average values of all the genotypes tested. A comparative analysis of the four clusters indicated a highly significant difference (df = 3, P < 0.001) among them for five of the six variables (soluble sugars: df = 3, Kruskal-Wallis chi-squared = 13.1, P = 0.004, Fig 4A; insoluble sugars: df = 3, F = 13.1, P = 0.01, Fig 4B; pH: df = 3, F = 17.7, P < 0.00001, Fig 4C; Flavonoids: df = 3, F = 20.95, P < 0.00001, Fig 4D and Phenols: df = 3, F = 9.11, P < 0.001, Fig 4E) that significantly described the clusters obtained. It was only tannins content that did not differ significantly (P>0.05) among the clusters. In general, clusters C1 and C2 had high soluble and insoluble sugars content whereas clusters C3 and C4 had high flavonoids and phenols content. Besides, cluster C4 exhibited an exceptionally high pH value compared to the other clusters.

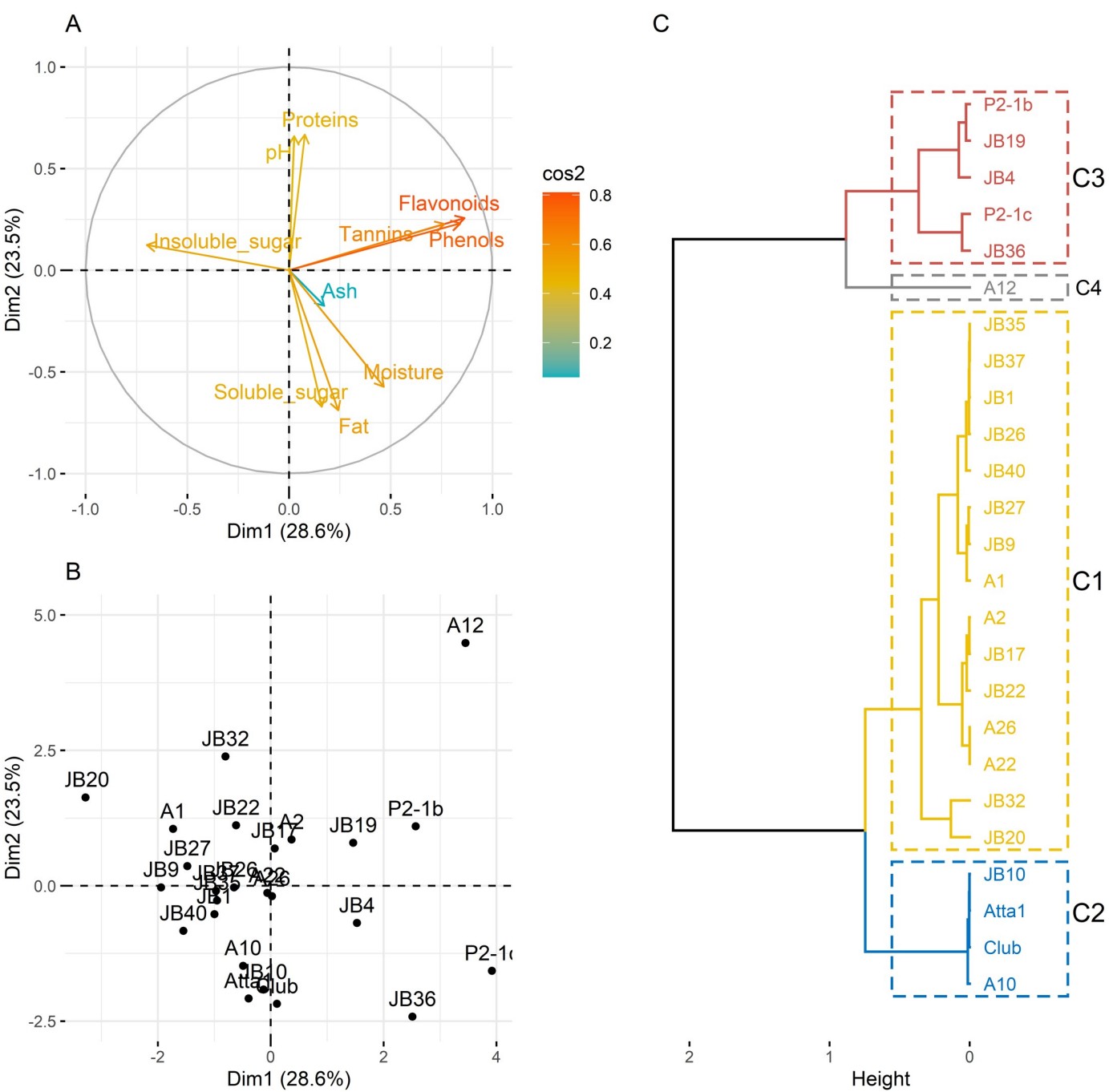

**Fig 3. Correlation circle (A), factor map (B) and dendrogramm illustrating the grouping of kola genotypes into clusters.**

## Discussion

Breeding fruits with enhanced nutritional and medicinal value is an important objective and has a major role to play in food and nutrition security and health of consumers especially in developing countries [58,59]. Clients along the kola value chain have already indicated preference for this trait and are demanding for it [32]. The objective of developing improved varieties of *C. nitida* with enhanced nutritional and pharmaceutical content is therefore aligned

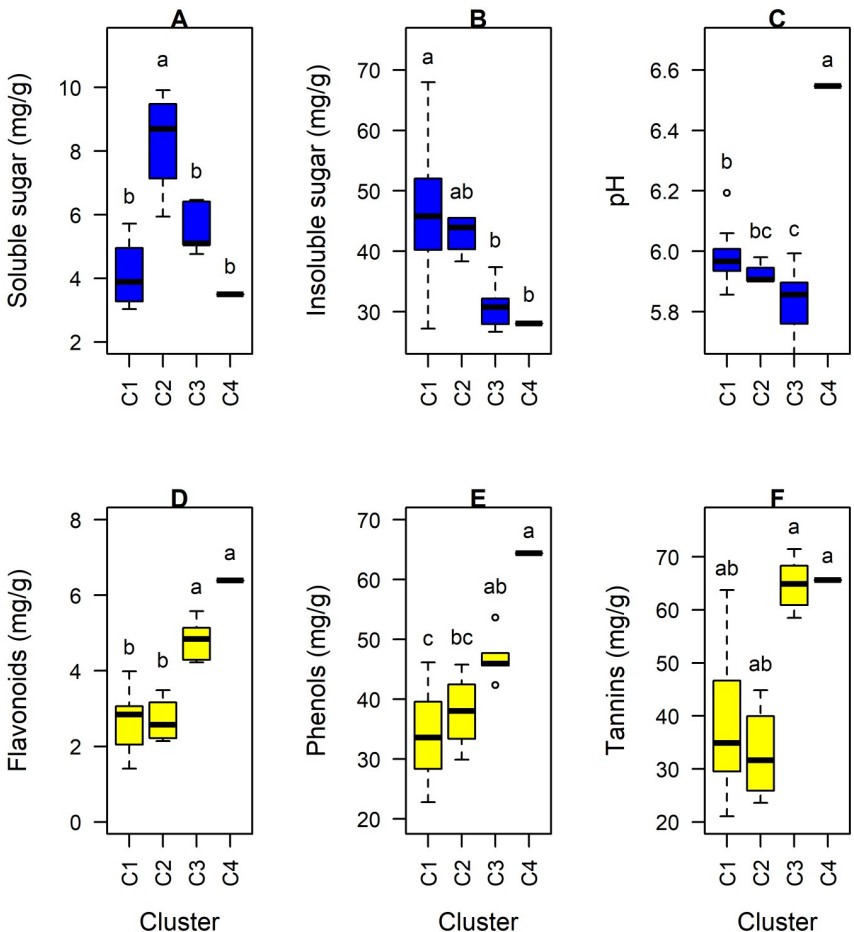

**Fig 4. Comparison of performance of the clusters using six characteristic variables.** Nutritional traits are in blue (A, B, C) and phenolic traits are in yellow (D, E, F).

towards a demand-led approach of breeding. Demand-led breeding approaches increase the likelihood of new varieties being adopted by farmers [60]. Consequently, there has been an upsurge in breeding for improved bio-compound contents in crops [61]. Although, bioactive compounds of kola nuts have been widely studied [8,20,62], knowledge on variability of kola genotypes for the contents of bioactive compounds is lacking. This study presents data on variation in nutritional (carbohydrates, proteins, ash, fats, moisture) and phenolic (polyphenols, flavonoids and tannins) contents in kola to characterize genotypic variability for selection and breeding purposes.

## Variation in nutritional and phenolic traits

The considerably varied CVs and significant differences observed in this study suggested that there is a variation among the 25 kola genotypes evaluated for the nutritional and phenolic traits. The high variability observed for these bio-active compounds provides opportunity to select promising genotypes for the improvement of nutraceutical contents [63]. Availability of high genetic variability is a pre-requisite to pragmatic identification and selection of desirable genotypes in plant breeding programmes [64]. The germplasm evaluated in this study

encompassed a high level of variation for all the nutritional and phenolic traits evaluated in this study.

The positive skewness coefficient for insoluble sugars, soluble sugars, flavonoids, pH, polyphenols, proteins and tannins indicated that the kola genotypes were inclined toward high contents of these traits. Ash, fat and moisture were negatively skewed suggesting their tended low content in the kola genotypes evaluated. This agrees with findings of Pursglove [8] who also reported low contents of fat, ash and moisture in kola. The low values of kurtosis for most of the traits except for ash and pH suggested that many of the kola genotypes were not near to the average and indicates a large number occuring on the extremes on either side.

## Estimates of genetic parameters of biocompounds in kola nuts

There were significant variations and high heritability estimates for the nutritional and phenolic parameters studied and this could facilitate phenotypic selection [65]. Selection could be based only on the phenotypic values observed due to the fact that genetic contribution was greater than that of the environment. Similar observations were reported by Girish et al. [66] and Falconer and Mackay [67]. Gerrano et al. [64] also showed a close difference between PCV and GCV values for elemental and nutrient contents of leaves of selected cowpea genotypes.

The high estimates of heritability and genetic gain for fat, insoluble sugars, soluble sugars, flavonoids, total phenols and tannins showed that selection for these traits will be very effective and reliable and are transferable to their progenies through breeding [68]. In the case of ash and pH, lower values of genetic gain was observed. This indicates that it will require many generations of crossings to accumulate the relevant genes/alleles for these traits. High genetic CV combined with high heritability estimates and genetic gain provide an indication that an expected amount of improvement through selection for the traits of interest is achievable [69]. Heritability is a fundamental parameter in genetics and allows a comparison of the relative importance of genes and environment to the variation of traits within and across populations. This important genetic parameter indicates the proportion of phenotypic variation that can be transferred to the next generation and indicates the extent to which a trait would respond to selection [67,70]. In addition, it gives an indication as to which extent a given trait will respond to selection [67]. For the nutritional and phenolic traits evaluated in this study, breeding methods based on progeny testing can be used to improve them. Achieving genetic advance drives improved germplasm and the release of new cultivars.

## Relationships between nutritional and phenolic traits of the 25 kola genotypes

The correlation between phenolic traits was positive and significant suggesting that they could be improved simultaneously. It also indicated that these phenolic traits can be independently targeted in a breeding programme if the other related traits does not give better grounds for discriminative selection [71]. The negative and insignificant association between nutritional and phenolic traits suggested that these traits should be improved independently.

Principal components analysis showed the contributions of the various components to total variation [72]. The contributions of each trait are indicated by the factor loadings. The loadings and eigenvectors indicate traits that are best for consideration in genetic improvement of a given crop. Flavonoids, total phenols, tannins, total proteins, soluble sugars and fats were characters that donated highly to the variation in the first two principal components which accounted for 52.13% of total variation. These traits are very important to discriminate kola

genotypes for nutrient and phenolic composition and deserve attention in breeding kola varieties with improved nutraceutical quality.

The results of PC1 and PC2 indicated that flavonoids, total phenols and tannins were well embodied on the factor map and thus deserve thoughtfulness in breeding improved varieties of kola. The top six genotypes that contributed high $CoS^2$ values were A12, P2-1b, JB36, JB20, JB32 and A10 suggesting they defined mainly PC 1 and PC 2 and would be important in selecting and breeding of kola cultivars with improved bioactive compounds content. Genotypes that had above the cut-off point are regarded very important for breeding for the traits of interest [73].

A cluster analysis is a good measure of diversity among and within crop species. It is able to group similar entries under one cluster [74]. The 25 kola genotypes were grouped into four separate clusters depending on the level of variation in bioactive compounds of the genotypes. The groupings of diversity and similarity among the kola genotypes observed in this study indicated possibility to identify and select desirable parents to create progenies with enhanced nutraceutical quality [75].

Genotype A12 was placed separately in a cluster. Such genotypes are denoted as singletons and are considered unique based on their performance in relation to traits of interest [76]. The kola genetic resources used in this study were collected from Asikem, Juaben and Tafo in Ghana with almost similar climatic conditions. This could explain why the genetic materials were not clustered on the basis of geographic origin. Nevertheless, the clustering indicates genotypic groups that are similar or have disimilar features and could be explored to identify individuals with desirable nutraceutical quality.

## Conclusion

Phenotypic variation in bioactive compounds content of twenty-five genotypes of kola was evaluated for the first time in Ghana. Significant and wide variations were found among the 25 kola genotypes for nutritional and phenolic traits. Although non-significant, correlations between nutritional traits and phenolic traits tended to be negative. In contrast, correlations among phenolic traits were all significant and positive. Phenolic traits exhibited higher heritability than nutritional traits. Based on the clustering, we suggested genotypes A12, JB9, JB19, JB32, P2-1b and P2-1c to be used to improve phenolic traits and the genotypes A10, Club, Atta1 and JB10 to improve nutritional traits. These genotypes could therefore be good candidates for use as parental lines to improve nutraceutical quality of kola for an enhanced utilization in food indsutries.

## Supporting information

**S1 Fig. Cutting-off plots for the study variables (A) and genotypes (B).** Variables and individuals cut by the red dashed lines are significantly represented on the the first two principal components.
(TIF)

**S1 Data. Raw data used in the statistical analysis.**
(CSV)

## Acknowledgments

Contributions of Mark Ofori, Foster Ansah, Emma Attah Yeboah, Abena Frempormaah, Edward Appiah in the harvesting and collection of pods from the field and Mrs Rafiatu Kotei and the technical team at the Biochemistry laboratory of CRIG in the analysis of the samples

are highly acknowledged. This paper is published with the permission of the Executive Director of the Cocoa Research Institute of Ghana as manuscript number CRIG/02/2020/048/005.

## Author Contributions

**Conceptualization:** Daniel Nyadanu.

**Data curation:** Daniel Nyadanu.

**Formal analysis:** Samuel Tetteh Lowor, Dèdéou Apocalypse Tchokponhoué.

**Investigation:** Daniel Nyadanu, Samuel Tetteh Lowor.

**Methodology:** Samuel Tetteh Lowor.

**Project administration:** Daniel Nyadanu.

**Resources:** Daniel Nyadanu, Samuel Tetteh Lowor, Jerome Agbesi Dogbatse.

**Software:** Dèdéou Apocalypse Tchokponhoué.

**Supervision:** Daniel Nyadanu, Micheal Brako-Marfo.

**Validation:** Daniel Nyadanu, Samuel Tetteh Lowor.

**Visualization:** Daniel Nyadanu, Prince Pobee.

**Writing – original draft:** Daniel Nyadanu.

**Writing – review & editing:** Daniel Nyadanu, Samuel Tetteh Lowor, Abraham Akpertey, Dèdéou Apocalypse Tchokponhoué, Prince Pobee, Jerome Agbesi Dogbatse, Daniel Okyere, Frederick Amon-Armah, Micheal Brako-Marfo.

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
