## [Editor Report · Decision Letter 0]

17 Jul 2020

PONE-D-20-17736

Genetic variability of bioactive compounds and selection for nutraceutical quality in kola [Cola nitida (Vent) Schott. and Endl.]

PLOS ONE

Dear Dr. Nyadanu,

Thank you for submitting your manuscript to PLOS ONE. After careful consideration, we have decided that your manuscript does not meet our criteria for publication and must therefore be rejected.

Specifically, the manuscript is about phenotypic variation for nutritional and phenolic traits in 25 genotypes of Cola nitida, a minor crop in Africa. Due to its cultivation in in very few countries in the world, it will be of limited interest for PLoS ONE readers. The literature presented in the manuscript omitted several studies in this crop. This study is too preliminary and does not advance science to the next level.  

I am sorry that we cannot be more positive on this occasion, but hope that you appreciate the reasons for this decision.

Yours sincerely,

Prasanta K. Subudhi, Ph.D.

Academic Editor

PLOS ONE

- - - - -

---

## [Author Response · Author response to Decision Letter 0]

19 Aug 2020

Response to Comment 1. We are particularly shocked about the idea that a paper is rejected because someone thought that the crop the manuscript targeted is a minor crop inAfrica. Sorry, but Minor crops in Africa also or should also matter in Science. In fact, PLoS ONE clearly indicates that: “it evaluates submitted manuscripts on the basis of methodological rigor and high ethical standards, regardless of perceived novelty”. Prof Preasanda K. Subuthi did not point out any methodological pitfall in this manuscript. We strongly believe that this decision needs to be reviewed since we are confident about the methodological rigour we used in the manuscript. We would be glad if the manuscript could be reassessed by another editor who truly knows the decision’s rules of the respected PLoS One Journal.

Editor’s comment 2. The literature presented in the manuscript omitted several studies in this crop

 Response to Comment 2. We would like to recall that this manuscript is not a review, so we only used literatures relevant to the specific aspects tackled in the manuscript. In addition, the editor comment seemed vague and did not point out specifically what he would like to see in the manuscript’s background. We would be happier if he indicated what he thought we did not take into account. We are confident that we used the last literature on the species Genetics and breeding.

Editor’s comment 3. This study is too preliminary and does not advance science to the next level. 

Response to Comment 3. Our study is the first in its kind to screen variability of bioactive compounds in Cola nitida. It offers first-order decision making information

---

## [Editor Report · Decision Letter 1]

16 Sep 2020

PONE-D-20-17736R1

Genetic variability of bioactive compounds and selection for nutraceutical quality in kola [Cola nitida (Vent) Schott. and Endl.]

PLOS ONE

Dear Dr. Nyadanu,

Thank you for submitting your manuscript to PLOS ONE. After careful consideration, we feel that it has merit but does not fully meet PLOS ONE’s publication criteria as it currently stands. Therefore, we invite you to submit a revised version of the manuscript that addresses the points raised during the review process.

ACADEMIC EDITOR: Authors are requested to make a major revision before resubmitting the manuscript for consideration.

We look forward to receiving your revised manuscript.

Kind regards,

Arun Jyoti Nath

Rosalyn B. Angeles-Shim, PhD

Academic Editors

PLOS ONE

Journal Requirements:

3. To comply with PLOS ONE submissions requirements for field studies, please provide the following information in the Methods section of the manuscript and in the “Ethics Statement” field of the submission form (via “Edit Submission”):

a) Provide the name of the authority who issued the permission for each location (for example, the authority responsible for a national park or other protected area of land or sea, the relevant regulatory body concerned with protection of wildlife, etc.). If the study was carried out on private land, please confirm that the owner of the land gave permission to conduct the study on this site.

b) For any locations/activities for which specific permission was not required, please

- i. state clearly that no specific permissions were required for these locations/activities, and provide details on why this is the case

- ii. confirm that the field studies did not involve endangered or protected species

For more information about PLOS ONE submissions requirements for field studies, please refer to http://journals.plos.org/plosone/s/submission-guidelines#loc-animal-research. 

Additional Editor Comments (if provided):

This manuscript entitled ‘Genetic variability of bioactive compounds and selection for nutraceutical quality in kola [Cola nitida (Vent) Schott. and Endl] by Nyadanu et al. reports the evaluation of 25 kola genotypes for nutritional traits and phenolic contents. Heritability estimates showed higher genetic control of the phenolic contents in kola nuts but not the nutritional characteristics. Genotypes that can be used as donors in breeding for both nutritional and phenolic traits were identified.

While the study provides a baseline for evaluating kola genotypes for breeding purposes and will be of interest to a range of scientists and breeders working on orphan crops, the manuscript needs to be extensively corrected for content (specifically materials and methodology), grammar, spelling, syntax and formatting before it can be considered for publication in PLoSONE.

1. Information on the experimental materials used in the study needs to be clarified. In page 5, line 92, the authors mentioned using 25 ‘clones’ of C. nitida. Clones are individuals having similar genetic constitution. For a study aimed at looking at genetic variability in any trait, the use of clones does not make any sense since each of these individuals will have the same genotype, ergo the same phenotype. Even if variation for a specific trait is observed, the variation would not be significant. But looking at Table 1, significant variation in pod yield, and nut weight and color among the ‘clones’ have been reported. While yield and weight are quantitative traits that are affected by the environment, the data presented in Table 1 does not conform with the expected intra-clonal variation. To make matters more complicated, in page 8, line 116, the authors mentioned ‘grafting’ in relation to the 25 ‘genotypes’ used in the study. It is unclear if the materials used in the

study were scions from a single plant (clones) grafted to 25 different rootstock genotypes or 25 different scions grafted to the same rootstock genotype? This is important especially because one of the objectives of the study is identifying genotypes that can be used in breeding. IF the materials used in the study are indeed grafts, the interaction between scion and grafts might have some effect in the traits examined and this should be determined prior to recommending genotypes that can be used as donors in breeding.

2. Page 4 line 67-73. The first and last sentence of this paragraph are contradictory.

3. Details about the methodology used for nitrogen, protein, carbohydrate and phenolic analysis should be provided. The readers should not have to go back to the cited papers for the full methodology used in the chemical analysis.

4. The manuscript needs to be checked for several grammatical, typographical and spelling errors, formatting, as well as syntax to improve the overall readability of the paper.

Examples:

a. Page 3, line 45: Cola acuminata should be changed to C. acuminate

b. Page 3, lines 49-50: This sentence is missing a period.

c. Page 3, line 51: “The kola industry offers’ not ‘offer’.

d. Page 3, line 59-60: Sentence can be paraphrased as ‘Bioactive compounds (phytonutrients) such as carotenoids and phenolic acids are health-promoting compounds that act against cardiovascular diseases and various types of cancer.’

e. Page 4, line 74: ‘bioactive compounds’ should be changed to ‘bioactive compound’

f. Page 4, line 74: wrong spelling of ‘content’ and ‘greatly’

g. Page 4, line 74: ‘…contents of fruits are’ not ‘…is’

h. Page 4, line 80-82: Sentence needs to be corrected. Wrong usage of grammar and sentence construction.

i. Page 5, line 99: Use the symbol for ‘degrees’ instead of superscripting the number ‘zero’.

j. Page 5, line 102: Remove comma after April.

k. Page 8, line 116: The phrase ‘The grafted materials the 25 kola genotypes..’ is unclear. Please paraphrase.

l. Page 11, line 169: Subheading title is unclear. Please paraphrase.

m. Page 11, line 171: wrong spelling of ‘between’

n. Page 11, lines 174, 176 and 177: What does the ‘()’ after the stat test mean?

o. Page 16, lines 259-261: the five variable mentioned need not be in bold. Also, flavonoids and phenols should not be capitalized.

p. Page 16, line 264: change ‘...flavonoids and phenols contents’ to ‘…flavonoid and phenol contents’.

q. Page 17 line 271: Remove ‘an’ in ‘Breeding fruits with an enhanced nutritional..’

r. Page 17, line 274: Change has to have

s. Page 17, line 283: Put a space between ‘..nutraceutical contents’ and the citation [48]

t. Page 21, line 322: No space after the sentence and the period.

u. Page 21, line 323: Add a space between ‘in’ and ‘a’.

v. The manner by which P values were written were not consistent across the manuscript.

w. Tables should be re-formatted according to PLoSONE specifications.

x. The references should be checked for formatting and typographical and spelling errors.

Note: The examples provided does not represent all the grammatical, typographical, spelling and formatting errors in the manuscript. The authors should have a responsibility in making sure that these issues are addressed before re-submitting.

Thank you.
---

## [Author Response · Author response to Decision Letter 1]

25 Oct 2020

Response to the reviewers

While the study provides a baseline for evaluating kola genotypes for breeding purposes and will be of interest to a range of scientists and breeders working on orphan crops, the manuscript needs to be extensively corrected for content (specifically materials and methods), grammar, spelling, syntax and formatting before it can be considered for publication in PloSONE.

Response: We wish to thank the editor for his positive comments and suggestions to fine tune the manuscript. The corrections suggested were accepted in good faith. Corrections have been carried out on the materials and methods, grammar, spelling, syntax and formatting as suggested. These corrections could be seen in the manuscript with the track changes. Thank you.

1. Information on the experimental materials used in the study needs to be clarified. In page 5, line 92, the authors mentioned using 25 clones of C. nitida. Clones are individuals having similar genetic constitution. For a study aimed at looking at genetic variability in any trait, the use of clones does not make any sense since each of these individuals will have the same genotype, ergo the same phenotype. Even if variation for a specific trait is observed, the variation would not be significant. But looking at Table 1, significant variation in pod yield and nut weight and color among the clones have been reported. While yield and weight are quantitative traits that are affected by the environment, the data presented in Table 1 does not conform to the expected intra-clonal variation. To make matters more complicated, in page 8, line 116, the authors mentioned ‘grafting’ in relation to the 25 genotypes used in the study. It is unclear if the materials used in the study were scions from a single plant (clones) grafted to 25 different rootstock genotypes or 25 different scions grafted to the same rootstock genotype? This is important especially because one of the objectives of the study is identifying genotypes that can be used in breeding. If the materials used in the study are indeed grafts, the interaction between scion and grafts might have some effect in the traits examined and this should be determined prior to recommending genotypes that can be used as donors in breeding.

Response

The 25 clones means 25 genotypes of kola. We refer to them as clones because they were vegetatively propagated through grafting and planted as a research trial. The 25 genotypes do not have the same genetic constitution and their phenotypes are not the same. The cuttings of the 25 genotypes were grafted on a rootstock of one genotype or of same genetic background. Therefore, there was no rootstock and scion interaction. To make it clear to our readers, the term “clones” has been changed to “genotypes” and the paragraphs rephrased to make issues clear to our readers. The changes can be seen clearly in the manuscript with track changes. Thank you.

2. Page 4 line 67-73. The first and last sentence of this paragraph are contradictory.

Response: Comment has been considered in revising the manuscript. The sentence has been rephrased to take out the contradiction.

3. Details about the methodology used for nitrogen, protein, carbohydrate and phenolic analysis should be provided. The readers should not have to go back to the cited papers for the full method.

Response: Details of the methodology for nitrogen, protein, carbohydrate and phenolic analysis has been provided in revising the manuscript. Thank you.

4. The manuscript needs to be checked for several grammatical, typographical and spelling errors, formatting, as well as syntax to improve the overall readability of the paper.

Response: This comment has been considered in revising the manuscript. The corrections on grammar, typographical and spelling errors, formatting and syntax was carried out throughout the manuscript.

Examples:

a. Page 3, line 45: Cola acuminate should be changed to C. acuminate

Response: The correct one is Cola acuminata. 

b. Page 3, lines 49-50: This sentence is missing a period.

Response:

c. Page 3, line 51: “The kola industry offers” not “offer”

 Response: Thank you for the correction. The correction has been implemented.

d. Page 3, line 59-60: Sentence can be paraphrased as “Bioactive compounds (phytonutrients) such as carotenoids and phenolic acids are health-promoting compounds that act against cardiovascular diseases and various types of cancer.”

Response: Thank you for the correction. The correction has been implemented.

e. Page 4, line 74: ‘bioactive compounds’ should be changed to ‘bioactive compound’

Response: Thank you for the correction. The correction has been implemented.

f. Page 4, line 74: wrong spelling of ‘content’ and ‘greatly’

Response: Thank you so much for identifying the spelling mistakes. Corrections has been implemented. They were rightly spelled.

g. Page 4, line 74: ‘ …contents of fruits are’ not ‘….is’

Response: Thank you for the correction. Correction has been implemented.

h. Page 4, line 80-82: Sentence needs to be corrected. Wrong usage of grammar and sentence correction.

Response: Thank you for pointing this out. The sentence has been rephrased and grammar corrected. This can be clearly seen in the manuscript with track changes.

 i.Page 5, line 99: Use the symbol for ‘degrees’ instead of superscripting the number ‘zero’

 Response: Thank you for the correction. The correction has been implemented. The symbol of “degrees” was used.

j. Page 5, line 102: Remove comma after April.

Response: Thank you for the correction. The correction has been carried out.

k. Page 8, line 116: The phrase “the grafted materials the 25 kola genotypes….’ Is unclear. Please paraphrase.

Response: Thank you for the correction. Those phrases has been paraphrased to make them clearer.

l. Page 11, line 169: Subheading title is unclear. Please paraphrase.

Response: Thank you for the correction. The subheading has been paraphrased to make it clear. The corrections can be clearly seen in the manuscript with track changes.

m. Page 11, line 171: wrong spelling of ‘between’

Response: Thank you for the correction. The Correction has been implemented. 

n. Page 11, lines 174, 176 and 177: What does the ‘()’ after the stat test mean?

Response: It is a script language which translates the way functions are written in R.

o. Page 16, lines 259-261: the five variable mentioned need not be in bold. Also, flavonoids and phenols should not be capitalized.

Response: Thank you for the correction. The corrections were considered in revising the manuscript.

p. Page 16, line 264: change ‘………flavonoids and phenols contents’ to ‘…..flavonoid and phenol contents’. 

Response: Thank you for the correction. The correction has been carried out.

q. Page 17 line 271: Remove ‘an’ in ‘Breeding fruits with an enhanced nutritional…’

Response: The correction was considered in revising the manuscript. Thank you.

r. Page 17, line 274: change has to have

Response: The correction was considered in revising the manuscript. Thank you.

s. Page 17, line 283: Put a space between ‘…nutraceutical contents’ and the citation [48]

Response: The correction was considered in revising the manuscript. Thank you.

t. Page 21, line 322: No space after the sentence and the period.

Response: The correction was considered in revising the manuscript. Thank you.

u. Page 21, line 323: Add a space between ‘in’ and ‘a’.

Response: The correction was considered in revising the manuscript. Thank you.

v. The manner by which P values were written were not consistent across the manuscript.

Response: The correction was considered in revising the manuscript. The manner in which P values are written is now uniform across the manuscript. Thank you.

w. Tables should be re-formatted according to PLoSONE specifications.

Response: Comment was considered in revising the manuscript. Tables has been formatted according to PLoSONE specifications.

x. The references should be checked for formatting and typographical and spelling errors.

Response: Comment was considered in revising the manuscript. All references were listed and well formatted. Typographical and spelling errors has been corrected.

The whole manuscript was revised and corrections were carried out. Corrections can be seen in the manuscript with track changes.

---

## [Editor Report · Decision Letter 2]

4 Nov 2020

PONE-D-20-17736R2

Genetic variability of bioactive compounds and selection for nutraceutical quality in kola [Cola nitida (Vent) Schott. and Endl.]

PLOS ONE

Dear Dr. Nyadanu,

Thank you for submitting your manuscript to PLOS ONE. After careful consideration, we feel that it has merit but does not fully meet PLOS ONE’s publication criteria as it currently stands. Therefore, we invite you to submit a revised version of the manuscript that addresses the points raised during the review process.

ACADEMIC EDITOR: Thank you for submitting the revised manuscript. Please provide a high resolution image for the Figure 2 and 3.

We look forward to receiving your revised manuscript.

Kind regards,

Arun Jyoti Nath

Academic Editor

PLOS ONE

---

## [Author Response · Author response to Decision Letter 2]

7 Nov 2020

Response to the reviewers

While the study provides a baseline for evaluating kola genotypes for breeding purposes and will be of interest to a range of scientists and breeders working on orphan crops, the manuscript needs to be extensively corrected for content (specifically materials and methods), grammar, spelling, syntax and formatting before it can be considered for publication in PloSONE.

Response: We wish to thank the editor for his positive comments and suggestions to fine tune the manuscript. The corrections suggested were accepted in good faith. Corrections have been carried out on the materials and methods, grammar, spelling, syntax and formatting as suggested. These corrections could be seen in the manuscript with the track changes. Thank you.

1. Information on the experimental materials used in the study needs to be clarified. In page 5, line 92, the authors mentioned using 25 clones of C. nitida. Clones are individuals having similar genetic constitution. For a study aimed at looking at genetic variability in any trait, the use of clones does not make any sense since each of these individuals will have the same genotype, ergo the same phenotype. Even if variation for a specific trait is observed, the variation would not be significant. But looking at Table 1, significant variation in pod yield and nut weight and color among the clones have been reported. While yield and weight are quantitative traits that are affected by the environment, the data presented in Table 1 does not conform to the expected intra-clonal variation. To make matters more complicated, in page 8, line 116, the authors mentioned ‘grafting’ in relation to the 25 genotypes used in the study. It is unclear if the materials used in the study were scions from a single plant (clones) grafted to 25 different rootstock genotypes or 25 different scions grafted to the same rootstock genotype? This is important especially because one of the objectives of the study is identifying genotypes that can be used in breeding. If the materials used in the study are indeed grafts, the interaction between scion and grafts might have some effect in the traits examined and this should be determined prior to recommending genotypes that can be used as donors in breeding.

Response

The 25 clones means 25 genotypes of kola. We refer to them as clones because they were vegetatively propagated through grafting and planted as a research trial. The 25 genotypes do not have the same genetic constitution and their phenotypes are not the same. The cuttings of the 25 genotypes were grafted on a rootstock of one genotype or of same genetic background. Therefore, there was no rootstock and scion interaction. To make it clear to our readers, the term “clones” has been changed to “genotypes” and the paragraphs rephrased to make issues clear to our readers. The changes can be seen clearly in the manuscript with track changes. Thank you.

2. Page 4 line 67-73. The first and last sentence of this paragraph are contradictory.

Response: Comment has been considered in revising the manuscript. The sentence has been rephrased to take out the contradiction.

3. Details about the methodology used for nitrogen, protein, carbohydrate and phenolic analysis should be provided. The readers should not have to go back to the cited papers for the full method.

Response: Details of the methodology for nitrogen, protein, carbohydrate and phenolic analysis has been provided in revising the manuscript. Thank you.

4. The manuscript needs to be checked for several grammatical, typographical and spelling errors, formatting, as well as syntax to improve the overall readability of the paper.

Response: This comment has been considered in revising the manuscript. The corrections on grammar, typographical and spelling errors, formatting and syntax was carried out throughout the manuscript.

Examples:

a. Page 3, line 45: Cola acuminate should be changed to C. acuminate

Response: The correct one is Cola acuminata. 

b. Page 3, lines 49-50: This sentence is missing a period.

Response:

c. Page 3, line 51: “The kola industry offers” not “offer”

 Response: Thank you for the correction. The correction has been implemented.

d. Page 3, line 59-60: Sentence can be paraphrased as “Bioactive compounds (phytonutrients) such as carotenoids and phenolic acids are health-promoting compounds that act against cardiovascular diseases and various types of cancer.”

Response: Thank you for the correction. The correction has been implemented.

e. Page 4, line 74: ‘bioactive compounds’ should be changed to ‘bioactive compound’

Response: Thank you for the correction. The correction has been implemented.

f. Page 4, line 74: wrong spelling of ‘content’ and ‘greatly’

Response: Thank you so much for identifying the spelling mistakes. Corrections has been implemented. They were rightly spelled.

g. Page 4, line 74: ‘ …contents of fruits are’ not ‘….is’

Response: Thank you for the correction. Correction has been implemented.

h. Page 4, line 80-82: Sentence needs to be corrected. Wrong usage of grammar and sentence correction.

Response: Thank you for pointing this out. The sentence has been rephrased and grammar corrected. This can be clearly seen in the manuscript with track changes.

 i.Page 5, line 99: Use the symbol for ‘degrees’ instead of superscripting the number ‘zero’

 Response: Thank you for the correction. The correction has been implemented. The symbol of “degrees” was used.

j. Page 5, line 102: Remove comma after April.

Response: Thank you for the correction. The correction has been carried out.

k. Page 8, line 116: The phrase “the grafted materials the 25 kola genotypes….’ Is unclear. Please paraphrase.

Response: Thank you for the correction. Those phrases has been paraphrased to make them clearer.

l. Page 11, line 169: Subheading title is unclear. Please paraphrase.

Response: Thank you for the correction. The subheading has been paraphrased to make it clear. The corrections can be clearly seen in the manuscript with track changes.

m. Page 11, line 171: wrong spelling of ‘between’

Response: Thank you for the correction. The Correction has been implemented. 

n. Page 11, lines 174, 176 and 177: What does the ‘()’ after the stat test mean?

Response: It is a script language which translates the way functions are written in R.

o. Page 16, lines 259-261: the five variable mentioned need not be in bold. Also, flavonoids and phenols should not be capitalized.

Response: Thank you for the correction. The corrections were considered in revising the manuscript.

p. Page 16, line 264: change ‘………flavonoids and phenols contents’ to ‘…..flavonoid and phenol contents’. 

Response: Thank you for the correction. The correction has been carried out.

q. Page 17 line 271: Remove ‘an’ in ‘Breeding fruits with an enhanced nutritional…’

Response: The correction was considered in revising the manuscript. Thank you.

r. Page 17, line 274: change has to have

Response: The correction was considered in revising the manuscript. Thank you.

s. Page 17, line 283: Put a space between ‘…nutraceutical contents’ and the citation [48]

Response: The correction was considered in revising the manuscript. Thank you.

t. Page 21, line 322: No space after the sentence and the period.

Response: The correction was considered in revising the manuscript. Thank you.

u. Page 21, line 323: Add a space between ‘in’ and ‘a’.

Response: The correction was considered in revising the manuscript. Thank you.

v. The manner by which P values were written were not consistent across the manuscript.

Response: The correction was considered in revising the manuscript. The manner in which P values are written is now uniform across the manuscript. Thank you.

w. Tables should be re-formatted according to PLoSONE specifications.

Response: Comment was considered in revising the manuscript. Tables has been formatted according to PLoSONE specifications.

x. The references should be checked for formatting and typographical and spelling errors.

Response: Comment was considered in revising the manuscript. All references were listed and well formatted. Typographical and spelling errors has been corrected.

The whole manuscript was revised and corrections were carried out. Corrections can be seen in the manuscript with track changes.

5. Please provide a high resolution for the Figure 2 and 3

Response: The resolution of the two figures were upgraded from a 300dpi resolution to >600dpi resolution. Figure 2 was divided into two parts (Figure 2-1 (A-F) and Figure 2-2 (G-J)) to be able to improve the resolution. This was used to effect changes in the figure number in the manuscript as well. The changes can be seen in the manuscript with the track changes.

---

## [Editor Report · Decision Letter 3]

13 Nov 2020

Genetic variability of bioactive compounds and selection for nutraceutical quality in kola [Cola nitida (Vent) Schott. and Endl.]

PONE-D-20-17736R3

Dear Dr. Nyadanu,

We’re pleased to inform you that your manuscript has been judged scientifically suitable for publication and will be formally accepted for publication once it meets all outstanding technical requirements.

Kind regards,

Arun Jyoti Nath

Academic Editor

PLOS ONE
---

## [Editor Report · Acceptance letter]

17 Nov 2020

PONE-D-20-17736R3 

Genetic variability of bioactive compounds and selection for nutraceutical quality in kola [*Cola nitida* (Vent) Schott. and Endl.] 

Dear Dr. Nyadanu:

I'm pleased to inform you that your manuscript has been deemed suitable for publication in PLOS ONE. Congratulations! Your manuscript is now with our production department. 

Kind regards, 

on behalf of

Dr. Arun Jyoti Nath 

Academic Editor

PLOS ONE